# Understanding the implementation of 'sick day guidance' to prevent acute kidney injury across a primary care setting in England: a qualitative evaluation

Anne-Marie Martindale,[1] Rebecca Elvey,[1] Susan J Howard,[1] Sheila McCorkindale,[2] Smeeta Sinha,[3] Tom Blakeman[1]

[1]National Institute for Health Research Collaboration for Leadership in Applied Health Research and Care, Salford Royal NHS Foundation Trust, Salford, UK
[2]NHS Salford Clinical Commissioning Group, Salford, UK
[3]Salford Royal NHS Foundation Trust, Salford, UK

**Correspondence to**
Dr Anne-Marie Martindale;
anne-marie.martindale@manchester.ac.uk

## ABSTRACT

**Objectives** The study sought to examine the implementation of sick day guidance cards designed to prevent acute kidney injury (AKI), in primary care settings.

**Design** Qualitative semistructured interviews were conducted and comparative analysis informed by normalisation process theory was undertaken to understand sense-making, implementation and appraisal of the cards and associated guidance.

**Setting** A single primary care health setting in the North of England.

**Participants** 29 participants took part in the qualitative evaluation: seven general practitioners, five practice nurses, five community pharmacists, four practice pharmacists, two administrators, one healthcare assistant and five patients.

**Intervention** The sick day guidance intervention was rolled out (2015–2016) in general practices (n=48) and community pharmacies (n=60). The materials consisted of a 'medicine sick day guidance' card, provided to patients who were taking the listed drugs. The card provided advice about medicines management during episodes of acute illness. An information leaflet was provided to healthcare practitioners and administrators suggesting how to use and give the cards.

**Results** Implementation of sick day guidance cards to prevent AKI entailed a new set of working practises across primary care. A tension existed between ensuring reach in administration of the cards to at risk populations while being confident to ensure patient understanding of their purpose and use. Communicating the concept of temporary cessation of medicines was a particular challenge and limited their administration to patient populations at higher risk of AKI, particularly those with less capacity to self-manage.

**Conclusions** Sick day guidance cards that focus solely on medicines management may be of limited patient benefit without adequate resourcing or if delivered as a standalone intervention. Development and evaluation of primary care interventions is urgently warranted to tackle the harm associated with AKI.

## INTRODUCTION

Addressing the harm related to acute kidney injury (AKI) is a worldwide priority.[1] AKI

### Strengths and limitations of this study

► Using normalisation process theory has allowed important insights to emerge into the comprehension, use and appraisal of the acute kidney injury (AKI) sick day card initiative.

► Interviews with a range of professionals (general practitioners, nurses, community and practice-based pharmacists, a healthcare assistant, practice administrators) and patients enhanced understanding of the individual and collective working practises surrounding the professional implementation AKI sick day guidance cards.

► Patient recruitment to the qualitative evaluation via general practice was slow and yielded only five patient-participants. This limited the analysis of patient use of sick day guidance in everyday life.

► Future study design would benefit from greater alignment between quantitative and qualitative elements of an evaluation.

is characterised as a sudden reduction in kidney function over hours or days.[2–4] It is a marker of illness severity and is seen as a 'force multiplier,' complicating episodes of acute illness.[3] As a clinical syndrome, the majority of cases of AKI are due to a combination of underlying infection, hypovolaemia (low circulatory blood volume), hypotension (low blood pressure) and medication effects.[3] Addressing these potentially modifiable factors are central to both the prevention and management of AKI and its associated burden.[2–4]

Across the UK, patient safety initiatives have been established to address the morbidity, mortality and costs linked to AKI.[2 5–7] In Scotland, informed by findings from a primary care study conducted by NHS Highland, medicine sick day rules have been made

available nationally through the Scottish Patient Safety Programme.[6 8] The introduction of medicine sick day rules relates to NHS Scotland Polypharmacy Guidance as well as national guidance, published by the National Institute for Health and Care Excellence (NICE) and by the Royal College of Physicians of Edinburgh UK.[4 9 10] These publications highlight a need to consider temporary cessation of medicines at times of acute illness.[4 9 10] That is, during these episodes, 'any drug that reduces blood pressure, circulating volume or renal blood flow' increases the risk of AKI.[3] Medicines that exacerbate this risk include non-steroidal anti-inflammatory drugs (NSAIDS), diuretics, ACE inhibitors and angiotensin II receptor blockers (ARBs).[3] In addition, the Scottish medicine sick day rules refer to the temporary cessation of metformin, which may accumulate at times of reduced kidney function, resulting in an increased risk of adverse effects.[6] The NHS Scotland 'Medicine Sick Day Rules' cards were developed through extraction of NHS Scotland Polypharmacy Guidance (2012) and were 'designed with input from pharmacists, doctors and patients'.[10 11] They provide instructions on temporarily stopping these specific types of medicines during episodes of acute illness.[6 8]

In England, within NHS England's Patient Safety Domain, the Think Kidneys Programme (https://www.thinkkidneys.nhs.uk) was established to tackle the harm associated with AKI.[12] Through the programme, resources have been developed for primary and secondary care, including an Interim Position Statement on 'Sick Day' Guidance, which highlights a clinical equipoise surrounding the systematic implementation of sick day guidance.[13]

It was in this wider context that a Clinical Commissioning Group (CCG), in partnership with the local hospital, embarked on service improvement initiatives to address the harm associated with AKI. Informed directly by the Scottish approach in conjunction with national guidance,[4 6 8] the CCG sought to implement the use of sick day guidance across general practices and community pharmacies within its boundaries. The Sick Day Guidance Project including an overview of the organisation of primary healthcare in England is outlined in table 1 as well as figures 1A,B. In accordance with NHS England Think Kidneys guidance, the project entailed formal evaluation. With a view to providing a platform for future larger scale evaluation, the study sought to explore and understand processes underpinning the implementation of sick day guidance in primary care.

## METHODS
### Study design
Aligned with the project objectives, normalisation process theory (NPT) provided a sensitising framework to inform the topic guide and explore the context, administration, interpretation and use of sick day guidance cards across a single primary healthcare setting in England.[14 15] NPT is a theory of implementation developed through an in-depth

analysis of chronic illness care in general practice.[14] It is a sociological theory that provides a structure to explore the individual and group work that people do surrounding the implementation of a complex intervention.[14–16]

### Data sampling
To explore the trajectory of implementation across the CCG, all general practices (n=48), community pharmacies (n=60) and practice-based pharmacists (n=4) involved in the project were invited to take part in the evaluation. Information packs were provided to explain what involvement entailed. To facilitate patient-participant engagement, general practices and community pharmacists were asked to provide information packs to patients who had received a card via a health practitioner. The final data sample of 29 interviews comprised: seven general practitioners (GPs), five practice nurses, five patients, five community pharmacists, four practice-based pharmacists, two managers (one medical practice manager and one community pharmacy manager) and a healthcare assistant, a person qualified to carry out routine healthcare tasks.

### Data collection
Two qualitative researchers (A-MM; RE) conducted the 29 semistructured interviews. These were conducted with participants across the CCG between June 2015 and April 2016. Participants received an approved participant information sheet and consent form via post or email. Both were read by the researcher prior to interview and participants had the opportunity to ask questions and have them answered satisfactorily. Informed consent was gained before each interview. Interviews with the GPs, practice nurses, administrators and the healthcare assistant took place in private locations within their general practices. Interviews with community pharmacists were also held at private locations at their places of work. Interviews with patients occurred at their homes. Interviews with three of the practice-based pharmacists took place at their place of work; one took place on the phone. The two researchers did not know any of the participants prior to interview. The interviews ranged in length from 9 to 66 minutes (median=33 min). They were digitally audio-recorded in compliance with participants' consent and professionally transcribed.

Interview topic guides were developed to explore the work being undertaken by professionals and patients surrounding the use of sick day guidance cards. NPT was used to inform the areas of questioning.[15] Topics for the health practitioners included previous knowledge of AKI and involvement in kidney health initiatives, their role in the intervention, sense-making and experiences of implementing and appraising the administration of sick day guidance cards. For patient-participant interviews, topics included: sense-making around health and illness, the context of card

**Table 1** The Sick Day Guidance Project TIDieR[39]

| TIDieR Item | Brief description |
|---|---|
| Name | Salford Kidney Implementation Project |
| (1) Why | The SPARC was established to ensure a shared strategy and optimise kidney care across the city.<br>The ambition of sick day guidance is to reduce the risk of avoidable harm to patients taking certain medications. Salford CCG in collaboration with SPARC defined the original implementation design of the sick day guidance intervention.<br>NIHR CLAHRC Greater Manchester works in partnership with Salford CCG to support implementation and evaluation of projects. NIHR CLAHRC Greater Manchester evaluated this CCG priority and supported the implementation of sick day guidance. |
| (2) What | Medicines sick day guidance was delivered in two phases of work. |
| (3) Materials | ▶ Sick day guidance cards that suggested the temporary cessation of medicines during bouts of sickness were produced, and the text was replicated from the NHS Highland sick day rules card.<br>▶ Two, one and a half hour, educational events were run for healthcare professionals, organised and delivered by the Steering Group. This included why AKI is important from a local and national context.<br>▶ Information leaflet outlining the sick day guidance project and guidance on how to use the sick day guidance cards and poster summarising this information for use in practice.<br>▶ Poster for patients promoting the sick day guidance card intervention to be used in waiting areas. |
| (4) Procedures | ▶ Training was offered to all GPs, practice nurses and the wider practice team and to community pharmacists for the sick day guidance card implementation.<br>▶ During Phase One, the cards were distributed to all community pharmacies and general practices accompanied by an information leaflet and poster with patient engagement instructions. Distribution was carried out by project facilitators face to face, to explain and address any questions arising.<br>▶ Two further face to face visits were made to each general practice and pharmacy by the NIHR CLAHRC Greater Manchester (GM) project team to reinforce the project/provide additional materials/support.<br>▶ The cards were to be provided to patients receiving the drugs listed on the card by general practices and community pharmacies.<br>▶ Posters were displayed in practice waiting areas promoting the intervention to patients.<br>▶ GPs and other practice staff were advised to record the intervention in Salford Integrated Records using Read code 80AG.<br>▶ During Phase Two, the practice-based pharmacists accessed patient health records from Salford Royal NHS Foundation Trust to identify those at risk of AKI and constructed a database to record relevant data.<br>▶ The practice-based pharmacists were to contact and educate patients on the sick day guidance and to issue a card. They were also expected to complete a medications review.<br>▶ Approval was sought to ensure the project was in keeping with national Think Kidneys guidance. |
| (5) Who | ▶ The NIHR CLAHRC GM project team (facilitation, project management and research staff).<br>▶ The Steering Group (clinical, pharmacist and managerial staff at Salford CCG and Salford Royal NHS Foundation Trust plus the NIHR CLAHRC GM project team).<br>▶ Salford CCG general practices and community pharmacies. |
| (6) How | The initial recruitment of GPs onto the project was implemented via email, and then three face to face visits were delivered per practice/pharmacy by NIHR CLAHRC GM project team to ensure full understanding of the sick day guidance project. Support was also gained from the local pharmaceutical committee. |

Continued

**Table 1** Continued

| TIDieR Item | Brief description |
|---|---|
| (7) Where | General practices (48) and community pharmacies (60) in Salford. 106 000 cards were provided to general practices and community pharmacies for administration to patients. |
| | In England, there were structural changes to the health service in 2013 and CCGs were formed. Each CCG covers the population of a defined area (ie, patients registered at general practices within the area) and is responsible for planning and commissioning the majority of health services in that area. Primary healthcare services are provided by GPs, community pharmacies, dentists and opticians. Patients register with a GP practice and attend that practice for appointments with a GP(s). Community pharmacies, also known as local chemist shops, are found on most local high streets, in shopping centres and also in many large supermarkets. Community pharmacies dispense prescription medicines, sell other (non-prescription) medicines and various other goods (typically health-related, baby and cosmetic products) and also provide other services, such as medicines use reviews. Patients do not register with a community pharmacy and may use any pharmacy (for dispensing or other services), although many patients become regular users of their local pharmacy. Pharmacists also work in general practices; such 'practice-based' pharmacists review medicines prescribing and take part in projects, such as the 'sick day guidance' intervention described here. |
| (8) When and how much | Cards were to be provided to a patient, when they attended a general practice appointment or visited a pharmacy between March 2015 and January 2016. |
| | Practice pharmacists contacted patients who fit within their criteria for being at risk of AKI. |
| (9) Tailoring | While guidance on the explanation to give patients (described above) was provided, professionals were expected to use their professional judgement in deciding how to deliver the intervention. |
| (10) Modifications | ▲ Opportunistic observations were conducted during facilitation visits. |
| | ▲ Cards were noticed on pharmacy counters, which were available for anyone visiting the pharmacy to pick up and take. |
| | ▲ Practice pharmacists encountered difficulties around the process of completing the record searches and communicating with patients in that there was not enough time to do this, consequently, no face to face appointments took place and pharmacists tried to contact patients by telephone. |
| | ▲ One practice pharmacist developed their own information sheet on patient with AKI that was posted out with cards. |
| (11) How well (planned) | Adherence and fidelity were not formally assessed; however, the facilitation visits were designed to provide flexible, ongoing support and advice on delivering the intervention and an understanding of how well the intervention was operating in practice was gained through these visits. |
| (12) How well (actual) | Practice pharmacists encountered barriers to obtaining the information they needed. |
| | ▲ CLAHRC facilitators gained understanding through their visits and the qualitative evaluation formally researched experiences of implementation — both these are documented in the CCG report. |
| | ▲ Sustained efforts had to be made to recruit health professionals and patients via medical practices. |

AKI, acute kidney injury; CCG, Clinical Commissioning Group; CLAHRC, Collaboration for Leadership in Applied Health Research and Care; NIHR, National Institute for Health Research; GP, general practitioner; SPARC, Salford Partnership for Advancing Renal Care.

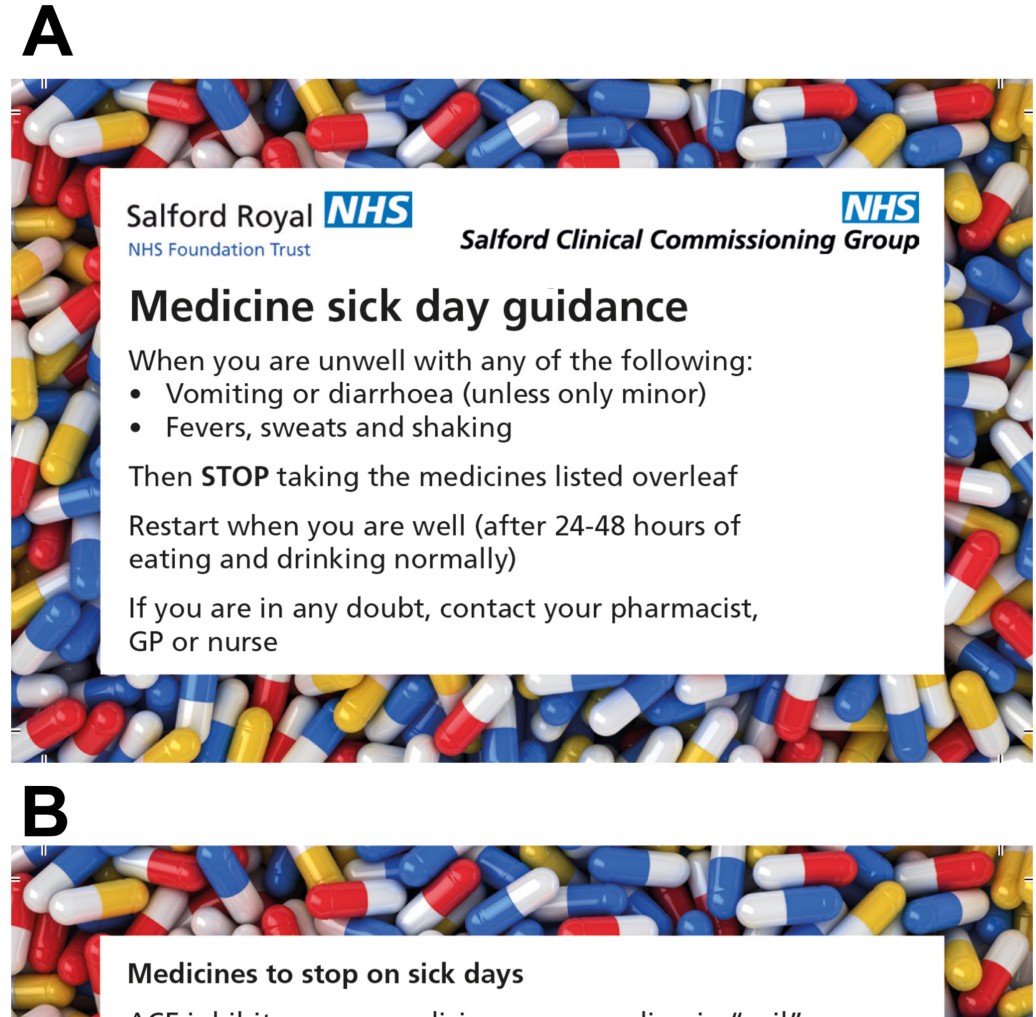

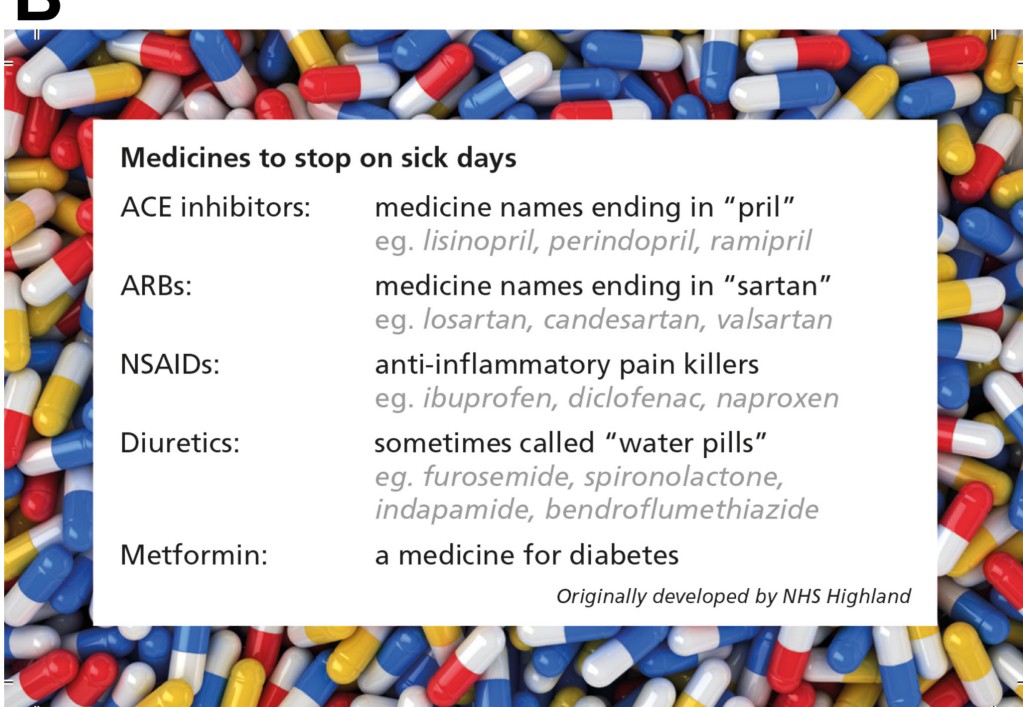

**Figure 1** (A) and (B) Sick day guidance card used during this project. The NHS Highland sick day rules card was reproduced with new logos.[6 8] ARB, angiotensin II receptor blockers; GP, general practitioner; NSAIDs, non-steroidal anti-inflammatory drugs.

giving and guidance explanation and comprehension and use of the guidance (table 2). Field notes about the encounter were written immediately after leaving the interview site and used to inform the analysis. Participants were asked if they wanted to receive a transcript post interview to check for accuracy, none did.

**Table 2** Summary Topic Guides for professional, managerial and support staff and patient interviews

| | | | |
|---|---|---|---|
| Health professionals, managerial and support staff | **Role in the AKI prevention project**<br>▶ Current role<br>▶ How it supported patients to prevent AKI before the project?<br>▶ Preparation for role in sick day guidance/AKI project<br>▶ Specific training/education<br>▶ Additional needs for training/education in the area of AKI prevention | **Views of the AKI prevention project**<br>▶ Who offered sick day rules/other AKI interventions to? (types of patients)<br>▶ How did you engage with patients<br>▶ What works well and why? (enablers)<br>▶ What does not work well and why? (barriers)<br>▶ Views of its impact on patients<br>▶ Views of the impact on your work and the rest of the healthcare team | **Integration with healthcare**<br>▶ How do sick day rules/other AKI initiatives, fit/link with other support for AKI prevention?<br>▶ Fit with long-term conditions management and other health needs and services?<br>▶ How do they fit/link with hospital care/social/voluntary sector?<br>▶ Contact/interaction with the rest of the primary healthcare team, secondary care team(s) around sick day guidance/AKI more generally?<br>▶ Which healthcare professionals are best placed to provide AKI prevention support? |
| Patients | **Context/history**<br>▶ Length of time of condition/taking medicines<br>▶ Perceptions of health and illness in everyday life<br>▶ Management of medicines and/or acute episodes of illness before the project (whether used a sick day guidance before/blister packs)<br>▶ Difficulties experienced around managing medicines and any needs? | **The sick day guidance/other kidney health interventions**<br>▶ How they found out about the service?<br>▶ Whether used the card or not?<br>▶ What do they find useful or like about it?<br>▶ What do they not find useful or dislike about it?<br>▶ Do they feel it has helped them? If so, how?<br>▶ Could it be improved? If so, how?<br>▶ Which healthcare practitioners could/should provide the cards? (where and when)<br>▶ Who are sick day cards/other AKI interventions suitable for? | **Coordination of care**<br>▶ Who is involved in their care?<br>▶ How/where does the sick day guidance/other support provided as part of the project, fit with other services or care received or other self-care undertaken? |

AKI, acute kidney injury.

## Data analysis

A-MM developed a thematic analysis framework using the evaluation objectives and the four core constructs of NPT to understand implementation.[14 15] NPT is concerned with social action rather than attitudes, and its four core constructs are coherence (sense-making), cognitive participation (relational work), collective action (operational work) and reflexive monitoring (appraisal).[14 15] The NPT constructs provided a pragmatic structure to consider different types of work surrounding the implementation of sick day guidance cards. Furthermore, it provided a sensitising framework to explore the relationships between different types of work being undertaken.[17] The questions asked of the health practitioner interview data included:

► how do they make sense of implementing the sick day card initiative? (coherence)
► what work have they done to implement the initiative? (operational work)
► how is the initiative being communicated or enacted by local others? (relational work)
► what judgments have been made about the initiative? (appraisal)

The questions we asked of the patient-participant data included:

► how does the participant make sense of health and illness? (coherence)
► what was the context of the participant receiving a card and guidance?
► how did they make sense of the card and implement the guidance in their day to day lives? (coherence, operational, relational work)
► how did they value the intervention? (appraisal).

As the interviews were completed and transcribed, data from each account were grouped according to role, which resulted in six datasets: GP, practice nurse and healthcare assistant, administration, community pharmacist, practice pharmacist and patient-participant. Thematic analysis using the transcripts, the audio recordings and the field notes was carried out by A-MM and TB. Each interview within a role group was analysed, and the findings were compared with those within the same group. Variations and similarities in context, sense-making, implementation and appraisal of the card were noted, explored and compared with the findings within and between role groups to enhance broader understanding.[18] Key themes and tensions underpinning implementation emerged through comparative, contextual analysis of individual and collective working practises underpinning introduction of sick day guidance cards.

## RESULTS

A version of the findings of this paper is included in a wider report that has been provided to the funding organisation.[19] AKI was viewed as a new phenomenon and the implementation of sick day guidance cards entailed a new set of working practises. Analysis indicated that AKI

prevention guidance was not necessarily a straightforward concept to understand or to communicate. Health practitioners thought the cards required some knowledge of illness symptoms and medicines and that patients had to decide how severe the symptoms were before acting or restarting their medication. One practice pharmacist stated:

'…patients don't understand what fever is…they think that if they've got a headache it's fever…we're trying to explain and they don't understand, or they say well, if I had a bout of diarrhoea do I stop the medication…it's severe. Well, what is severe, you know? Obviously it's very subjective…' (SKHIP13PP).

Comparative analysis highlighted a tension between the need to achieve reach to the populations deemed at risk (ie, those taking medicines specified on the card) and at the same time ensure comprehension concerning use of the guidance. There was evidence that this tension influenced the implementation of the sick day guidance intervention. The following sections describe the different approaches employed.

### Administration of the sick day guidance card in conjunction with face-to-face communication

A common theme was health professionals and patients valuing the need to explain the guidance in person. One patient reflected:

'I don't think that it should be just put on a counter… I don't think, number one, they'll read it, number two, they'll digest what's on it, or number three, they'll apply it to themselves' (SKHIP22PA).

A practice nurse thought dialogue was also important to reduce miscommunication, avoid patient confusion and additional GP workload:

'I always explain …There's no point giving someone a card if they don't understand what it's for…my grandma wouldn't understand that. She'd probably misinterpret that and…stop taking everything' (SKHIP25PN).

Analysis of health practitioner and patient accounts revealed that patients responded to the guidance in a variety of ways, not always as intended. One patient participant used the terms sickness and illness interchangeably and spoke of different classifications of illness. She asked which type the guidance card was referring to, to be confident of following the instructions properly:

'What do you define as illness…? Well, I suppose I don't know… I've got arthritis, that's not an illness it's just a thing of life when you get older…I've had spinal surgery, but they're not illnesses…' (SKHIP22PA).

Two health practitioners reported instances of patients with medication-associated diarrhoea stopping their tablets since receiving a card. This unintended consequence of the initiative lead to those patients being prescribed alternative medication to alleviate the side

effect. A couple of patient-participant accounts revealed a lack of willingness to follow the guidance as it had not been implemented by their hospital specialist, whose opinion they trusted, and they did not want to make their condition worse:

'*I'd rather feel sick than have a problem with the high blood pressure…*' (SKHIP31PA).

The concept of temporary cessation of medicines required careful consideration, for example when to stop, restart and what dosage to reinstate:

'*We don't have enough data or…best practice… if you stop the metformin or whatever medication how long do you stop it for…? Then after a week are you going to restart them again on the ten milligram or are you going to start them on the 1.5, the 2.5…?*' (SKHIP14GP).

Although valued by the health practitioners interviewed, implementation of sick day guidance initiative demanded extra work. In general practice, this was deemed less problematic when it fitted into existing long-term condition review appointments, particularly with practice nurses or healthcare assistants. In community pharmacies, implementation sat more readily within face-to-face medication review appointments or opportunistic over-the-counter interactions, including the purchase of NSAIDS such as ibuprofen. One community pharmacist used the purchase of antidiarrhoeal or sickness medications as an opportunity to administer AKI guidance:

'*…when people have been coming in to buy stuff for sickness or diarrhoea… If it turns out that they're on one of the medications that's on the card, then we'll give them a card then as well and explain about it*' (SKHIP5CP).

There were limits to the implementation of sick day guidance in patient populations deemed at increased risk of AKI. Concerns were expressed across the health professionals interviewed that the cards and temporary cessation of medications were not suitable for patients with cognitive impairments such as Alzheimer's disease, reduced literacy in English, those with advanced learning difficulties or visual impairments or for elderly housebound patients taking multiple medicines. One community pharmacist commented on the difficulties facing patients and carers using dosette box (blister pack) systems:

'*they (patients) might have four or five tiny little white ones, and then if they're elderly or they can't see the markings, they don't know what tablet they should be stopping…. if it was a family member looking out for it, that would be I guess possible, but a lot of the carers are not allowed to alter any medication*' (SKHIP7CP).

## Administration of sick day guidance cards to patients in conjunction with telephone consultations

Phase Two of the project entailed Practice Pharmacists supporting the implementation of the sick day guidance

cards in general practices (see table 1). All of the four CCG employed pharmacists valued and engaged with the project. However, they outlined difficulties fitting the implementation in with their pre-existing workload. There were more patients to work with than anticipated, and the searches, writing to patients, communicating with them and feeding the results back to GPs took longer to complete than the pharmacists described having time for.

To implement the project in this context, a decision was made to have telephone conversations with patients rather than face-to-face interactions. However, this created additional challenges. The phone calls took as long as the face-to-face encounters as the pharmacists expressed a professional need to do things '*properly*'. They reported patients not always being happy to talk with a perceived stranger on the phone about their health. Patient understanding was harder to assess and patients did not necessarily agree to enact the guidance if they became ill. Unlike the face-to-face GP and practice nurse consultations, patients on the other end of the phone had no prior trusting relationship with the practice pharmacist. One pharmacist tried to mitigate some of these issues by talking with a GP in advance of phoning:

'*…I'm not going to just pick up the phone and ring this patient now, I'm going to ask the GP what he thinks… for the slightly elderly- some patients, perhaps mental health issues.…They obviously know their patients much better than I do so I always take their advice*' (SKHIP11PP).

The community pharmacists also spoke of the difficulties of assessing patient comprehension in this way:

'*I've had to phone patients …if you've got a query or the prescription will be changed or we'll want to question something …sometimes they're on the ball, they completely know, and sometimes they're just so confused*' (SKHIP7CP).

## Sick day guidance cards being administered without verbal or written communication

Instructions administered to health practitioners (figure 2) stressed the need for dialogue with patients to check understanding. However, accounts indicated that this did not always occur. Reasons included other work demands during a practice-based consultation, limited time for dialogue, forgetting to discuss it and some lack of confidence about what to say, partly because of the limited evidence base and so as not to confuse patients, especially those who were less fluent in English:

'*we have quite a lot of different ethnicities here…they've got limited English I think they're not quite sure and it takes quite a while explaining …about what medicines to stop, when to stop it, when to restart it…*' (SKHIP10PN).

Though the community pharmacists were willing to talk with patients about the guidance cards, time shortages and other work demands impinged on implementation. One community pharmacist stated:

## MEDICINES AND DEHYDRATION: SICK DAY GUIDANCE

### Offer the following information at the time of giving the card

- Some medicines shouldn't be taken when you have an illness that makes you dehydrated. This is because they can either increase the risk of dehydration or because dehydration can lead to potentially serious side effects of the medicine.
- The medicine you are taking that falls into this category is [tell patient which medicine].
- Illnesses that can cause dehydration are vomiting, diarrhoea and fever.
- This advice does not apply to minor sickness or diarrhoea, which means a single episode.
- If your medicines are in a blister pack you must take it to the chemists so the chemist can show you which ones you need to stop.
- If you have heart failure you may stop these medicines for a maximum of 48 hours but after that you need to contact your GP or heart failure team for further advice.

### The list of medicines on the card is not exhaustive but they are highlighted because:

- diuretics can cause dehydration or make dehydration more likely in an ill patient;
- ACE inhibitors, angiotensin II receptor blockers and NSAIDs may impair kidney function in a dehydrated patient, which could lead to kidney failure;
- metformin dehydration increases the risk of lactic acidosis, a serious and potentially life-threatening side effect of metformin.

**Figure 2** Guidance provided to health practitioners (shortened form). GP, general practitioner; NSAIDs, non-steroidal anti-inflammatory drugs.

'*Half the time it's remembering to do it because you're thinking about that many different things*' (SKHIP5CP).

In addition, they did not always have face-to-face contact with patients:

'*we've got like 900 of our own patients and we just make the packs and then send them out and delivery, so we don't actually have that much patient contact*' (SKHIP7CP).

Some health practitioners felt that the cards were self-explanatory. One practice nurse said:

'*vomiting is vomiting and diarrhoea is diarrhoea*' (SKHIP25PN).

However, others did not agree. One GP thought it was really important to provide patients with written material to aid understanding and compliance:

'*with certain other sort of medicine regimes, we ask them to stop temporarily if there's a drug interaction and patients are okay with that, as long as you give them sort of written instructions and they know exactly why they're stopping. A lot of it is to do with the understanding. They don't like stopping things if they don't understand why…*' (SKHIP20GP).

A couple of patient accounts referred to finding cards in public information areas of medical practices and community pharmacies. One patient who found a card in this manner wanted to share the sick day guidance message:

'*…I went into the pharmacy last week, they were on the counter…I picked one up and brought it home …I think it's such a good idea that I've given one to my sister*' (SKHIP22PA).

### Communication of AKI risk, but limited use of a sick day guidance card

One GP worked exclusively with patients in care homes across the CCG, which included patients who were diagnosed with cognition limiting conditions such as dementia. Though the guidance messages were deemed pertinent to these groups of patients more vulnerable to AKI, their use was limited due to a potential lack of understanding:

'*So we have the card. We didn't use it a lot…We used it to give to the carers. I used it to give to a few of the patients that have capacity*' (SKHIP14GP).

The need for appropriate training for carers, nursing staff and associated social workers was raised, beyond the level of the sick day guidance card. Specifically there was felt to be an ongoing need for health practitioners to highlight the importance of fluid management in conjunction with medicines management:

'…*they* (dementia patients) *ended up not eating or drinking, worsening of the renal function and become unwell and they end up in hospital…' So it's working with the carer as well to understand…. It's serious things that they might die from, not being hydrated'* (SKHIP14GP).

## DISCUSSION
### Principal findings
Implementation of sick day guidance cards to prevent community based AKI entailed a new set of working practises. The temporary cessation of medicines during episodes of acute illness was not necessarily a straightforward concept to understand or communicate. Comparative analysis of participants' accounts highlighted a tension between ensuring reach in administration of the cards to at risk populations while being confident to ensure patient understanding of their purpose and use.

### Strengths and weaknesses of this study
Unlike an earlier study,[20] a key strength of this evaluation was to conduct an in-depth exploration of systematic rollout across a single healthcare setting. The study was hypothesis generating, and use of NPT provided a sensitising framework for data collection and analysis.[14–16] Recognising that all theories have the potential to structure and constrain analysis, NPT was chosen as it ensured that a range of individual and collective working practises were considered during analysis.[14 15] Methods to enhance the trustworthiness of the findings, including their transferability, entailed exploring types of work undertaken in both general practices and community pharmacies as well as their use by a range of health professionals in these different settings.[21]

The study entailed comparative analysis of both patient and professional accounts in order to explore their use in clinical interactions as well as in everyday life. Thematic analysis has illuminated a key tension between achieving reach while ensuring comprehension of the card and its instructions. However, a larger sample size might have resulted in the identification of additional themes that may have had an impact on this theoretical framework. Further research is required to enhance patient understanding and use. Professional accounts allowed descriptions of experiences of use by patients, though difficulties were encountered recruiting patient-participants who had experiences of having used a sick day guidance card at times of acute illness. It is important to acknowledge that only five patients were interviewed in spite of extensive recruitment efforts. It is not possible to determine how many patients received information packs as we did not ask practices to keep a record, to reduce work load. Health professionals did not always pass on the evaluation recruitment packs to patients, and the patients we interviewed had not used the cards to date, which could help to explain limited patient involvement. Workload pressures were cited as reasons for health professionals declining to participate in the evaluation.

During the course of the interviews, health practitioners were asked about patient sense-making, use and appraisal of the guidance cards. In light of limited patient involvement, these accounts became more important. We acknowledge that they are third order interpretations; our interpretations of what health practitioners reported about patients' sense-making, appraisal and use of the cards. However, the comparative approach taken has facilitated understanding of the pluralistic journeys of the cards and their intended and unintended messages and trajectories from card giver to patient across the 29 interviews. Future studies may benefit from sampling patients who have been coded in general practice as having been provided sick day guidance (ie, Read Code 8OAG. 'Provision of information about AKI'[22] and also who have been coded with an episode of acute illness (eg, gastroenteritis, acute respiratory infection). In doing so, this this would enable purposeful sampling according to medical history including evidence of multimorbidity. As stated in the CCG report, 106 000 cards (see table 1) were distributed across general practices and community pharmacies within the time frame of the project.[19] However, community pharmacists were not required to record administration to patients and inaccuracies in coding in general practice limited the potential for a robust quantitative analysis. Future study design would benefit from greater alignment between quantitative and qualitative elements of an evaluation.[19]

### Comparison with other studies
In terms of professional responsibility, there are recognised boundaries to the role of GPs in supporting self-management.[23] The findings of this study resonate and build on the results of previous research, which highlighted issues around the consistency of clinical message and the additional work required to reduce the risk of harm from AKI using medicines management interventions.[20 24] The intervention was conducted at a time when concern was raised that UK general practice workload may be at 'saturation point.'[25] Results suggested that this influenced engagement with the CCG-led initiative.

Though currently available through the Scottish Patient Safety Programme,[6] the findings from this qualitative study resonate with recently published literature, which highlights a need for a more robust evidence base surrounding both the implementation and effectiveness of sick day guidance cards.[26–28] A recent systematic review showed that 'there is no evidence of the impact of drug cessation interventions on AKI incidence during intercurrent illness in primary or secondary care.'[26 28] In terms of implementation, studies evaluating AKI interventions in secondary care indicate that establishing clinician approval is critical with a need for intervention design to take into account 'how technologies, people and organisations dynamically interact' in order for AKI interventions to become integrated into routine clinical practice.[29 30] Interventions that disrupt workflow 'may not

be sustainable even if there has been a positive impact on care.'[29]

Results from a population-based cohort study indicate that patient comorbidities including chronic kidney disease are much more strongly associated with AKI and that treatment with either an ACE Inhibitor or an ARB is only associated with a small increase in AKI risk.[27] That is, younger patients with limited comorbidity (eg, on ACEI for treatment of hypertension) have a low absolute risk of AKI, while patients living with multimorbidity in whom there may be professional concerns about ensuring effective risk communication, have a much higher risk of AKI.[27]

### Implications for clinicians, policy makers and future research

In the UK, NICE recommends raising awareness of AKI in higher risk population groups with specific reference to patients who: have existing CKD; have had a previous episode of illness complicated by AKI and/or have neurological or cognitive impairment and who may be reliant on carers for support with fluid intake during an acute illness (eg, those with cognitive impairment).[31] This may help address a knowledge gap in patient and public understanding of the importance in the maintenance of kidney health. A survey conducted in 2014 on behalf of NHS England indicated that 'about half of the population in Great Britain do not think their kidneys make urine' and 'only an eighth (12%) of interviewees thought their kidneys had a role in processing medicines.'[32] However, the findings from this study suggest an evidence base is urgently warranted to determine how best to resource effective self-management support for higher risk patient populations. Targeting patients who have had an episode of illness complicated by AKI may be particularly important. As a marker of vulnerability, data from a Welsh study showed that around 50% of their patient population died within 14 months; the study also revealed high rates of hospital readmission.[33] Of the 733 patients discharged following a hospital admission complicated by AKI, there were 498 rehospitalisation events in a 6-month period.[33]

The NHS England Urgent and Emergency Care Review also emphasised the need for better support for people to self-care.[34] Our analysis in conjunction with the research by Mansfield et al[27] suggests sick day guidance cards alone, that focus solely on temporary cessation of medicines, are unlikely to be sufficient to reduce the harm associated with AKI. The CCG chose to implement the Scottish (NHS Highland) Medicine Sick Day Rules card without significant modification of content or format.[6] However, the current intervention may need modifying, to make it suitable for use with various populations, such as provision in languages other than English. For example, recognising the risks of the 'triple whammy' combination of NSAIDS prescribed in conjunction with diuretics and renin–angiotensin system inhibitors (ie, ACE inhibitors and ARBs), is there potential for misunderstanding if NSAIDS are included in a sick day guidance card administered to patients with heart failure?[35] Both usability testing and experience-based codesign are methodological approaches that may optimise the development of an intervention that takes into account patient and carer experience.[36] The findings suggest other strategies may need to be resourced to prevent AKI in people with complex health and social care needs such as those living with dementia. A key issue raised was to provide better education and support for carers (both professional and informal). The Royal College of General Practitioners has provided guidance on the development of 'carer friendly' practises and the establishment of Patient Participation Groups may be a mechanism to resource and integrate support for carers into the organisation of acute care.[37 38]

## CONCLUSION

The findings from this qualitative evaluation suggest that there are boundaries to the implementation of sick day guidance cards to prevent acute kidney injury in primary care. A common theme was the need to ensure patient understanding of their purpose and use. Communicating the concept of temporary cessation of medicines was a particular challenge and limited their administration to patient populations at higher risk of AKI, particularly those with less capacity to self-manage. The analysis suggests that sick day guidance cards that focus solely on medicines management may be of limited benefit without either adequate resourcing or if delivered as a standalone intervention. Development and evaluation of a primary care intervention encompassing a range of initiatives to tackle the harm associated with AKI is warranted.

**Acknowledgements** The authors would like to thank the project steering group for their input and guidance throughout the study.

**Contributors** TB, RE, SJH, SM and SS contributed to the conception and design of the work. RE and A-MM collected the data, RE, A-MM and TB analysed the data. A-MM, TB, RE and SJH drafted the manuscript and all authors contributed to interpreting the data, revising the work for intellectual content, agree to be accountable for the work and have approved the manuscript.

**Funding** This project was funded by the National Institute for Health Research Collaboration for Leadership in Applied Health Research and Care (NIHR CLAHRC) Greater Manchester and NHS Salford CCG. The NIHR CLAHRC Greater Manchester is a partnership between providers and commissioners from the NHS, industry and the third sector as well as clinical and research staff from the University of Manchester.

**Disclaimer** The views expressed in this article are those of the authors and not necessarily those of the NHS, NIHR or the Department of Health.

**Competing interests** None declared.

**Ethics approval** Ethical approval was gained from Leeds West Research Ethics Committee (REC Reference Number: 15/YH/0174). Informed consent was gained from all participants prior to interview.

**Provenance and peer review** Not commissioned; externally peer reviewed.

**Data sharing statement** The data have been stored securely with password protected files to ensure confidentiality, in keeping with the research protocol and good data management guidelines. It will not be shared.

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
