## [Reviewer comments · BMJ Open]

ARTICLE DETAILS

TITLE (PROVISIONAL)	Understanding the implementation of 'sick day guidance' to prevent Acute Kidney Injury across a primary care setting in England: a qualitative evaluation
AUTHORS	Martindale, Anne-Marie; Elvey, Rebecca; Howard, Susan; McCorkindale, Sheila; Sinha, Smeeta; Blakeman, Tom

VERSION 1 – REVIEW

REVIEWER	Jill Vanmassenhove Ghent University Hospital, Belgium
REVIEW RETURNED	17-May-2017

GENERAL COMMENTS	This is a well written paper concerning a very relevant subject, which is the prevention of community acquired AKI I have some comments -In my opinion it would be useful to include a small paragraph on the organization of the health care system in England (compared to other countries) so that readers can judge for themselves how to compare results to their own setting. E.g. the concept of "community pharmacy" is something that does not exist in the country where I live.-How and why was this particular primary health care practice chosen? Is it reasonable to expect the same outcome in other practices across England? Although authors do mention the small number (of especially patients) in this research project, I still feel they need to underpin more why they think this small number does not rule out a reliable evaluation of the 'sick days rule' concept.-What were the reasons so many patients did not want to do the interview? What were the characteristics of the patients who participated (age? Comorbidities?) And why did not more GP's and pharmacists participate in doing the interview?-How many patients were addressed in total? I would think that addressing all patients on the drugs mentioned in the pamphlet would increase the workload for caregivers tremendously. It might be more interesting to select patients according to comorbidities such as CKD, diabetes and heart failure and previous AKI.-I think you should collaborate more on the need for communication between hospital specialists and GP's, especially in case of a previous AKI episode during hospitalization.-It might make more sense also to focus on the GP's as the preferred caregiver to explain the sick days rule concept to the patients and consider the pharmacist as the stakeholder with a secondary role, repeating/reinforcing the GP's advice.
---

	This would avoid that patients get conflicting information or get information from a party they trust less. Again, this also relates to how the health care system is organized. If patients have to get their repeat prescriptions every three months via the GP it seems appropriate that the GP takes the 'leadership' in this project since patients would not turn up at the pharmacist with a prescription without having seen their GP first. -Authors should also address the fact that not all patients with prescriptions for potentially toxic drugs, will actually take these drugs. -I'm also not keen on having the anti-inflammatory drugs on that list because it suggests that they are safe outside episodes of acute illnesses. However, we can expect that part of these patients who are taking diuretics and RAAS blockers will have CKD and/or heart failure; conditions where you want to avoid to ever use anti-inflammatory drugs in the first place. -Shouldn't radiological procedures with administration of contrast iodide and surgery also be considered on this pamphlet? Although I agree there is no hard evidence either and it is also not the goal of this study to criticize the original pamphlet.. -Different ethnicities and language issues seem to play a role in implementation. Again, these are factors that might fluctuate heavily in between countries. -Did the pharmacists and GP's involved have previous experience with the sick day rules concept?
--	--

REVIEWER	Professor Nicola Thomas London South Bank University
REVIEW RETURNED	18-May-2017

GENERAL COMMENTS	 1. The abstract is clear and concise and Table 1 is extremely useful in providing details of the programme. Check your spelling of practice/practise throughout as there are some typos. 2. There is no description of NPT prior to the data analysis section, so please include 1-2 sentences about what this is exactly in the methods section, otherwise the reader is left wondering what this theory is and why it has been used. 3. I would like to see further discussion of the use of the NHS Highland cards. Did you use them without any review of whether they were 'fit for purpose'? So for example was the font size and wording suitable for older people? If they were not scrutinised then the reason why needs to be included in the discussion. 4. Page 11 -what/who is a 'health care assistant account'? 5. There is no mention of where the interview questions came from and this needs an additional section. Did the questions come from the lit review, from expert opinion etc?? 6. Why did you use NPT to analyse the data? Why not other data analysis methods? There needs to be discussion of this. 7. It is not clear in the Results section how the sub-headings relate to the NPT analysis. Are the sub-headings in bold the ones that you identified? You need to explain this in the introductory paragraph of the Results section.
--

	8. It is usual to put the 'Comparison with other studies' section at the beginning of the Discussion, so I would move this section to be first. 9. In the limitations section I would like to see some commentary of whether the cards should have been reviewed prior to use. eg. did NHS Highland pilot them, did you pilot them, did you involve patients in their development etc.If not, then this needs to be a recommendation. 10. Consent for publication. If the findings have already been published online, is there a copyright issue with this? Overall an interesting study but further discussion in parts would improve the paper, both in terms of understanding of methodology/methods and also implications for future practice.
--	---

REVIEWER	Suren Kanagasundaram Newcastle upon Tyne Hospitals NHS Foundation Trust, UK Institute of Cellular Medicine, Newcastle University, UK
REVIEW RETURNED	21-May-2017

GENERAL COMMENTS	The authors present a timely evaluation of sick day guidance implementation across a single CCG. Their qualitative evaluation, conducted within the framework of Normalisation Process Theory, has revealed a tension between the need for 'reach' across the at risk population and a patient / carer-centred focus which is felt to be necessary for successful implementation of a potentially complex self-management programme. The paper's value lies in its insights into an intervention for which the evidence base is limited but which, nevertheless, has an understandable momentum from the national imperative to reduce the harm arising from preventable AKI. The study is important but might be enhanced by considering the following: Most importantly, and as acknowledged, the patient and carer viewpoint is limited, diminishing the study's core messages. Similarly, although a significant number of staff interviews have been undertaken, these can't necessarily be taken as an homogeneous whole given the differing roles each group has in the intervention. It seems unlikely that thematic saturation has been achieved within all these groups although this hasn't been mentioned in the paper. Further, the qualitative output might have been easier to interpret with a quantitative context. For instance, can the authors give some idea of the number of patients and carers receiving the intervention (at least as logged by Read code)? Could they have been described, demographically and in terms of their care requirements? Also, how was AKI risk determined? AKI risk factor profiling tools are not well validated and may identify a large target population without discrimination for those at highest risk. Implementation and integration may have quite different approaches based on size and nature of the target group of patients.
---

	Finally, some qualitative examination of hospital-based AKI QI initiatives has been undertaken: Clinical Kidney Journal 2016 doi: 10.1093/ckj/sfw054. , and our own work in, Clinical Kidney Journal 2015;9(1):57-62 doi: 10.1093/ckj/sfv130. ...it may be useful, but certainly not essential, to comment on issues of end-user acceptance and intervention credibility raised in both references and alluded to in the present paper.
--	---

VERSION 1 – AUTHOR RESPONSE

Reviewer 1 Jill Vanmassenhove

Comment:

In my opinion it would be useful to include a small paragraph on the organization of the health care system in England (compared to other countries) so that readers can judge for themselves how to compare results to their own setting. E.g. the concept of “community pharmacy” is something that does not exist in the country where I live’.

Response:

An Overview of the organisation of the health care system in England is now provided in Table 1. The introduction has been amended on page 6-7 to signpost the reader to this table.

Comment:

‘How and why was this particular primary health care practice chosen? Is it reasonable to expect the same outcome in other practices across England?’

NIHR Collaboration for Leadership in Applied Health Research and Care Greater Manchester works in partnership with Salford CCG to support implementation and evaluation of projects. NIHR CLAHRC Greater Manchester supported the evaluation of this CCG priority and approach to the implementation of sick day guidance. Table 1 has been amended to make this more explicit.

Rather than seeking to evaluate outcomes, through the use of qualitative methods, this study was hypothesis generating. On page 7 we have stated that the study sought to explore and understand processes underpinning the implementation of sick day guidance in primary care. The trustworthiness of the findings including the transferability of findings is critiqued in the Strengths and Weaknesses section of the discussion. Amendments have been made to this section on pages 24 and 25 to make this more explicit.

Comment:

‘Although authors do mention the small number (of especially patients) in this research project, I still feel they need to underpin more why they think this small number does not rule out a reliable evaluation of the ‘sick days rule’ concept’.

Response:

We have amended the Strengths and Weaknesses section of the Discussion on pages 24-26 to make it clearer that patient interviews were of value to consider the trajectory, understanding and use of the card through primary care, but that further recruitment would have enhanced the trustworthiness of the study.

Comment:

'What were the reasons so many patients did not want to do the interview?

The Strengths and Weaknesses section of the Discussion on page 24-25 has been expanded to address this point.

Comment:

What were the characteristics of the patients who participated (age? Comorbidities?)

Response:

We do not have demographic details of the patient participants. This is acknowledged in the Strengths and Weaknesses section on page 25-26 in terms of improving patient recruitment including a need for purposeful sampling according to the presence of multi-morbidity.

Comment:

And why did not more GP's and pharmacists participate in doing the interview?

Response:

Workload pressures were cited as reasons for health professionals declining to participate in the evaluation. This is acknowledged on page 25 of the manuscript. We did not remunerate for participation in the evaluation.

Comment:

'How many patients were addressed in total? I would think that addressing all patients on the drugs mentioned in the pamphlet would increase the workload for caregivers tremendously. It might be more interesting to select patients according to comorbidities such as CKD, diabetes and heart failure and previous AKI'.

Response:

We were not able to accurately capture how many patients received a sick day guidance card. Addressing this methodological limitation is discussed on page 26.

A key finding was the tension between achieving reach in terms of administration whilst ensuring effective communication and comprehension. This is highlighted in the results and discussion sections.

The CCG decided to implement the Scottish/NHS Highland approach. The introduction and the discussion (see page 28) have been amended to make this more explicit. Table 1 also states that the design of the cards was a replication of the Scottish/NHS Highland Medicine Sick Day Rules. The evaluation focused on exploring issues surrounding the approach chosen.

Comment:

'I think you should collaborate more on the need for communication between hospital specialists and GP's, especially in case of a previous AKI episode during hospitalization'.

Response:

As mentioned, the CCG decided to implement the Scottish/NHS Highland approach. The suggested approach to focus on patients who have had an episode of illness complicated by AKI is now raised in the discussion on page 28. References have also been updated to support these amendments.

Comment:

'It might make more sense also to focus on the GP's as the preferred caregiver to explain the sick days rule concept to the patients and consider the pharmacist as the stakeholder with a secondary role, repeating/reinforcing the GP's advice. This would avoid that patients get conflicting information or get information from a party they trust less. Again, this also relates to how the health care system is organized. If patients have to get their repeat prescriptions every three months via the GP it seems appropriate that the GP takes the 'leadership' in this project since patients would not turn up at the pharmacist with a prescription without having seen their GP first'.

Response:

As mentioned above, the CCG decided to implement sick day guidance cards based on the Scottish/NHS Highland approach. The evaluation focused on exploring the approach chosen.

Comment:

'Authors should also address the fact that not all patients with prescriptions for potentially toxic drugs, will actually take these drugs.'

Response:

'I'm also not keen on having the anti-inflammatory drugs on that list because it suggests that they are safe outside episodes of acute illnesses. However, we can expect that part of these patients who are taking diuretics and RAAS blockers will have CKD and/or heart failure; conditions where you want to avoid to ever use anti-inflammatory drugs in the first place. -Shouldn't radiological procedures with administration of contrast iodide and surgery also be considered on this pamphlet? Although I agree there is no hard evidence either and it is also not the goal of this study to criticize the original pamphlet'.

The CCG decided to implement the Scottish/NHS Highland approach – the introduction has been amended on pages 6 and 7 to make it more explicit on how the card were first designed by NHS Highland. The Discussion has also been amended to highlight the issue raised concerning the inclusion of NSAIDS on the sick day guidance cards. Please see page 28.

Comment:

'Different ethnicities and language issues seem to play a role in implementation. Again, these are factors that might fluctuate heavily in between countries'.

The discussion has been amended to acknowledge this this point. Please see page 28.

Comment:

'Did the pharmacists and GP's involved have previous experience with the sick day rules concept?'

Response:

This time bounded CCG project was the first systematic roll out of sick day guidance cards in Salford CCG. To our best knowledge were not aware of previous use of sick day guidance cards.

Reviewer 2 Professor Nicola Thomas

'Thank you for asking me to review this paper. It is an interesting study and an important topic for primary care and renal teams. I hope my suggestions below are useful'.

'The abstract is clear and concise and Table 1 is extremely useful in providing details of the programme. Check your spelling of practice/practise throughout as there are some typos'.
We have checked the manuscript and amended accordingly.

Comment:

'There is no description of NPT prior to the data analysis section, so please include 1-2 sentences about what this is exactly in the methods section, otherwise the reader is left wondering what this theory is and why it has been used'.

Response:

Sentences have been added to the Study design section in Methods, page 12, in order to provide an overview of NPT and its relevance to this study.

Comment:

'I would like to see further discussion of the use of the NHS Highland cards. Did you use them without any review of whether they were 'fit for purpose'? So for example was the font size and wording suitable for older people? If they were not scrutinised then the reason why it needs to be included in the discussion'.

Response:

The CCG decided to implement the Scottish/NHS Highland approach – the introduction on pages 6 and 7 has been amended to make it more explicit on how the cards were first designed by NHS Highland. The evaluation explored factors underpinning the approach chosen.

Comment:

'Page 11 -what/who is a 'health care assistant account?'

Response:

We have amended the text where it says 'health care assistant' on page 12.

Comment:

'There is no mention of where the interview questions came from and this needs an additional section. Did the questions come from the lit review, from expert opinion etc.?'

Response:

We have made amendments to the study design section (page 12) and data collection (page 13) to make this more explicit. Please also see the response to the next question below.

Comment:

'Why did you use NPT to analyse the data? Why not other data analysis methods? There needs to be discussion of this'.

Response:

The first paragraph in the Strengths and Weaknesses section of the Discussion has been amended to address this point – please see page 24-25. The study design section on page 12 has also been amended to provide an overview of NPT and its relevance to this study. NPT provided a theoretical lens to enable us to consider findings from different angles, individual sense-making, individual work, and sense-making and work in conjunction with others in a primary care environment.

Comment:

'It is not clear in the Results section how the sub-headings relate to the NPT analysis. Are the sub-headings in bold these that you identified? You need to explain this in the introductory paragraph of the Results section'.

Response:

Findings emerged through comparative analysis of the different types of work being undertaken surrounding the introduction of sick day guidance cards. NPT provided a sensitising framework to explore the different types of work being undertaken and the relationship between them. The Methods section (pages 12-15) has been amended to make this more explicit. As such, the findings and tension that emerged are not categorised according to the NPT constructs. That is, NPT was used as sensitising tool rather than a rigid framework.

Comment:

'It is usual to put the 'Comparison with other studies' section at the beginning of the Discussion, so I would move this section to be first'.

Response:

We would be grateful for guidance from the editorial team concerning the structure of the Discussion section. The current structure was based on a previous paper published in BMJ Open (see Blakeman et al, 2016. BMJ Open 2016;6:e012865. doi:10.1136/bmjopen-2016-012865).

Comment:

'In the limitations section I would like to see some commentary of whether the cards should have been reviewed prior to use. e.g. did NHS Highland pilot them, did you pilot them, did you involve patients in their development etc. If not, then this needs to be a recommendation'.

Response:

The CCG decided to implement the Scottish/NHS Highland approach – the introduction has been amended (see pages 6 and 7) to make it more explicit on how the card was first designed by NHS Highland. The evaluation explored factors underpinning the approach chosen by the CCG. As recommended, future work to optimise the design of a complex 'sick day guidance' intervention is raised in the discussion (see page 28).

Comment:

'Consent for publication. If the findings have already been published online, is there a copyright issue with this?'

Response:

We took guidance from a colleague who is a Deputy Editor in Chief of a Health Service Journal on this matter, which informed a decision for the CCG to publish its report in a timely manner and that the CCG report then needs to be explicitly referenced in a subsequent peer review publication. The findings have not been published in any other peer-review publication. We would be grateful if the editorial team could help clarify the reviewer's query.

Comment:

'Overall an interesting study but further discussion in parts would improve the paper, both in terms of understanding of methodology/methods and also implications for future practice'.

We have made amendments to the introduction, methods and discussion sections in response to the reviewers' helpful comments.

Reviewer 3 Suren Kanagasundaram

'Thanks for the opportunity to review this submission. The authors present a timely evaluation of sick day guidance implementation across a single CCG. Their qualitative evaluation, conducted within the framework of Normalisation Process Theory, has revealed a tension between the need for 'reach' across the at risk population and a patient / carer-centred focus which is felt to be necessary for successful implementation of a potentially complex self-management programme. The paper's value lies in its insights into an intervention for which the evidence base is limited but which, nevertheless, has an understandable momentum from the national imperative to reduce the harm arising from preventable AKI.

Comment:

The study is important but might be enhanced by considering the following: Most importantly, and as acknowledged, the patient and carer viewpoint is limited, diminishing the study's core messages. Similarly, although a significant number of staff interviews have been undertaken, these can't necessarily be taken as a homogeneous whole given the differing roles each group has in the intervention.

It seems unlikely that thematic saturation has been achieved within all these groups although this hasn't been mentioned in the paper'.

Response:

The Strengths and Weaknesses section in the Discussion has been amended to acknowledge this methodological issue. Please see pages 24-26.

Comment:

'Further, the qualitative output might have been easier to interpret with a quantitative context. For instance, can the authors give some idea of the number of patients and carers receiving the intervention (at least as logged by Read code)? Could they have been described, demographically and in terms of their care requirements?'

Response:

The Strengths and Weaknesses section of the Discussion has been amended to address this point. Please see pages 24 to 26. A quantitative evaluation of the intervention was planned, however, it became apparent that coding at general practices varied, in terms of levels of coding (not all instances of the provision of cards were coded) and also that different codes were used across some practices. This is discussed in the CCG Report (see Reference 20). In addition, the provision of cards at community pharmacies was not recorded; patients do not register with pharmacies and consultations such as those where the cards were distributed are not routinely recorded in community pharmacies. As stated in the discussion this limited the potential for an accurate quantitative evaluation. The strengths and limitations bullet points on page 5 have also been amended to highlight this methodological issue.

Comment:

'Also, how was AKI risk determined? AKI risk factor profiling tools are not well validated and may identify a large target population without discrimination for those at highest risk. Implementation and integration may have quite different approaches based on size and nature of the target group of patients'.

Response:

The evaluation explored factors underpinning the approach chosen by the CCG. The evaluation reports by NHS Highland state that the Medicine Sick Day rule cards 'were designed through extraction of NHS Scotland Polypharmacy Guidance and entailed input from pharmacists, doctors and patients.' The CCG decided to implement the Scottish/NHS Highland approach.

The introduction has been amended on pages 6 and 7 and two references 6 and 8 have been added to make this more explicit. The issue of which patient population to target is raised in the discussion and in doing so, acknowledges the current limited evidence based as well as papers by Mansfield et al (reference 28) and also Wonnacott et al (reference 34) highlighting who might be a greater risk.

Comment:

'Finally, some qualitative examination of hospital-based AKI QI initiatives has been undertaken: Clinical Kidney Journal 2016 doi: 10.1093/ckj/sfw054. , and our own work in, Clinical Kidney Journal 2015;9(1):57-62 doi: 10.1093/ckj/sfv130. ...it may be useful, but certainly not essential, to comment on issues of end-user acceptance and intervention credibility raised in both references and alluded to in the present paper'.

Response:

Thank you for these helpful papers and suggestion, the discussion has been amended. Please see pages 27-28.

Finally, could we request that the 3rd author has her name spelled Susan J Howard please, as the J has been missed out of the step 4 on-line submission, and it won't change. Thank-you

VERSION 2 – REVIEW

REVIEWER	Jill Vanmassenhove Ghent University Hospital Belgium
REVIEW RETURNED	15-Jul-2017

GENERAL COMMENTS	The authors have addressed my concerns appropriately and i therefore have no further comments
---

REVIEWER	Prof Nicola Thomas London South Bank University, UK
REVIEW RETURNED	23-Jul-2017

GENERAL COMMENTS	Thank you very much for sending this paper to me again for further review. Thank you to the authors for sending a clear explanation of the changes that have been made. The paper is much improved and clearer for the reader to follow, The COREQ checklist has also been completed, so I am very happy to now recommend acceptance.
---

REVIEWER	Suren Kanagasundaram Renal Services, Newcastle upon Tyne Hospitals NHS Foundation Trust, UK Institute of Cellular Medicine, Newcastle University, UK
REVIEW RETURNED	26-Jul-2017

GENERAL COMMENTS	Thanks for the invitation to score this revised manuscript and to the authors for their helpful responses, largely addressing the concerns raised in the initial review. As already stated, the insights from this paper offer an important opportunity for reflection before similar QI programmes are rolled out, elsewhere, as well as a new launch point for more extensive qualitative evaluations of similar interventions. Where concerns still exist, I think these are relatively minor. Firstly, the authors may wish to re-word the following statement (under “Strengths and Limitations” of the study; page 5) that the study has allowed a “comprehensive understanding of the sense making, use and approach of the AKI sick day card initiative”. An alternative wording might be that it has allowed, “important insights into”. To expand - my understanding of the differences between thematic and theoretical saturation are that the former occurs when no new themes are emerging in data analysis, whilst the latter only occurs when the inter-connectivity between the revealed themes is fully described – hence requiring more extensive recruitment. Unfortunately, it seems unlikely that even thematic saturation has occurred given the limited numbers in individual staff groups and in numbers of patients, recruited. I’d recommend qualifying the statement that “theoretical saturation” is said to have been achieved at least “in terms of illuminating the key tension between achieving reach whilst ensuring comprehension” (2nd sentence, 2nd paragraph, page 25) with a caveat that the limited recruitment may have prevented additional themes from emerging that may have had an impact on this theoretical framework. A further caveat is needed, I think, on the merging of disparate staff groups into more homogeneous wholes for the purpose of analysis. In addition, response rates can be inferred from the manuscript as being modest (with 24 staff interviewees identified after approaching 48 practices and 60 community pharmacies) or extremely low (with 5 patients interviewed after distribution of over 100,000 sick day guidance cards to healthcare providers). A more explicit statement on actual response rate would be useful – i.e. how many invitations to participate were sent out? I hope the authors find these comments helpful and look forward to hearing their view.
---

VERSION 2 – AUTHOR RESPONSE

Reviewer 3

'Thanks for the invitation to score this revised manuscript and to the authors for their helpful responses, largely addressing the concerns raised in the initial review. As already stated, the insights from this paper offer an important opportunity for reflection before similar QI programmes are rolled out, elsewhere, as well as a new launch point for more extensive qualitative evaluations of similar interventions. Where concerns still exist, I think these are relatively minor'.

Comment:

'Firstly, the authors may wish to re-word the following statement (under "Strengths and Limitations" of the study; page 5) that the study has allowed a "comprehensive understanding of the sense making, use and approach of the AKI sick day card initiative". An alternative wording might be that it has allowed, "important insights into".

Response:

The authors have re-worded the first statement under the 'Strengths and Limitations' heading on page 5 to read

'Using Normalisation Process Theory (NPT) has allowed important insights to emerge into the comprehension, use and appraisal of the AKI sick day card initiative'.

'To expand - my understanding of the differences between thematic and theoretical saturation are that the former occurs when no new themes are emerging in data analysis, whilst the latter only occurs when the inter-connectivity between the revealed themes is fully described – hence requiring more extensive recruitment.

Comment:

Unfortunately, it seems unlikely that even thematic saturation has occurred given the limited numbers in individual staff groups and in numbers of patients, recruited. I'd recommend qualifying the statement that "theoretical saturation" is said to have been achieved at least "in terms of illuminating the key tension between achieving reach whilst ensuring comprehension" (2nd sentence, 2nd paragraph, page 25) with a caveat that the limited recruitment may have prevented additional themes from emerging that may have had an impact on this theoretical framework.

Response:

The following caveat has now been added to page 24, second paragraph, second and third sentence 'Thematic analysis has illuminated a key tension between achieving reach whilst ensuring comprehension of the card and its instructions. However, a larger sample size might have resulted in the identification of additional themes that may have had an impact on this theoretical framework. Further research is required to enhance patient understanding and use'.

Comment:

A further caveat is needed, I think, on the merging of disparate staff groups into more homogeneous wholes for the purpose of analysis'.

Response:

The aim of the research was to make sense of individual, then role based comprehension, use and evaluation of the card. Groups were not treated as homogenous wholes. The text has been amended to make this clearer. Page 16, third sentence, first paragraph...

'Each interview within a role group was analysed, and the findings were compared with those within the same group. Variations and similarities in context, sense-making, implementation and appraisal of the card were noted, explored and compared with the findings within and between role groups to enhance broader understanding. 19 Key themes and tensions underpinning implementation emerged through comparative, contextual analysis of individual and collective working practises underpinning introduction of sick day guidance cards'.

'In addition, response rates can be inferred from the manuscript as being modest (with 24 staff interviewees identified after approaching 48 practices and 60 community pharmacies) or extremely low (with 5 patients interviewed after distribution of over 100,000 sick day guidance cards to healthcare providers). A more explicit statement on actual response rate would be useful – i.e. how many invitations to participate were sent out?'

The authors are unable to provide this information, as practices were not asked to keep records of how many patients they had informed, to reduce work. The text has been amended to reflect this. Page 25, under the Strengths and Weaknesses of this study sub heading towards the end of the second paragraph- 'It is not possible to determine how many patients received information packs as we did not ask practices to keep a record, to reduce work load'.

We hope we have addressed Reviewer 3's comments in full and look forward to hearing from you.

VERSION 3 – REVIEW

REVIEWER	Suren Kanagasundaram Newcastle upon Tyne Hospitals NHS Foundation Trust, UK Newcastle University, UK
REVIEW RETURNED	23-Aug-2017
GENERAL COMMENTS	Many thanks. My further queries have been fully addressed.